# Advanced Ti–Nb–Ta Alloys for Bone Implants with Improved Functionality

**DOI:** 10.3390/jfb15020046

**Published:** 2024-02-17

**Authors:** Jan-Oliver Sass, Marie-Luise Sellin, Elisa Kauertz, Jan Johannsen, Markus Weinmann, Melanie Stenzel, Marcus Frank, Danny Vogel, Rainer Bader, Anika Jonitz-Heincke

**Affiliations:** 1Research Laboratory for Biomechanics and Implant Technology, Department of Orthopaedics, Rostock University Medical Center, Doberaner Straße 142, 18057 Rostock, Germany; marie-luise.sellin@med.uni-rostock.de (M.-L.S.); elisa.kauertz@uni-rostock.de (E.K.); danny.vogel@med.uni-rostock.de (D.V.); rainer.bader@med.uni-rostock.de (R.B.); anika.jonitz-heincke@med.uni-rostock.de (A.J.-H.); 2Fraunhofer Research Institution for Additive Manufacturing Technologies IAPT, Am Schleusengraben 14, 21029 Hamburg, Germany; jan.johannsen@iapt.fraunhofer.de; 3TANIOBIS GmbH, Im Schleeke 78-91, 38642 Goslar, Germany; markus.weinmann@taniobis.com (M.W.); melanie.stenzel@taniobis.com (M.S.); 4Medical Biology and Electron Microscopy Center, Rostock University Medical Center, Strempelstraße 14, 18057 Rostock, Germany; marcus.frank@med.uni-rostock.de; 5Department Life, Light and Matter, University of Rostock, 18051 Rostock, Germany

**Keywords:** implant material, β-titanium alloy, mechanical properties, biological properties, laser beam powder bed fusion

## Abstract

The additive manufacturing of titanium–niobium–tantalum alloys with nominal chemical compositions Ti–xNb–6Ta (x = 20, 27, 35) by means of laser beam powder bed fusion is reported, and their potential as implant materials is elaborated by mechanical and biological characterization. The properties of dense specimens manufactured in different build orientations and of open porous Ti–20Nb–6Ta specimens are evaluated. Compression tests indicate that strength and elasticity are influenced by the chemical composition and build orientation. The minimum elasticity is always observed in the 90° orientation. It is lowest for Ti–20Nb–6Ta (43.2 ± 2.7 GPa) and can be further reduced to 8.1 ± 1.0 GPa for open porous specimens (*p* < 0.001). Furthermore, human osteoblasts are cultivated for 7 and 14 days on as-printed specimens and their biological response is compared to that of Ti–6Al–4V. Build orientation and cultivation time significantly affect the gene expression profile of osteogenic differentiation markers. Incomplete cell spreading is observed in specimens manufactured in 0° build orientation, whereas widely stretched cells are observed in 90° build orientation, i.e., parallel to the build direction. Compared to Ti–6Al–4V, Ti–Nb–Ta specimens promote improved osteogenesis and reduce the induction of inflammation. Accordingly, Ti–xNb–6Ta alloys have favorable mechanical and biological properties with great potential for application in orthopedic implants.

## 1. Introduction

Adequate mechanical properties, high corrosion resistance, biocompatibility, and sufficient osteogenesis are required for implant materials in direct bone contact [1]. Implants in direct contact with bone made of Ti–6Al–4V (values are always given in wt. %) have shown good long-term results in clinical use [2,3], though there is growing concern about potential damage to human cells from released aluminum and vanadium ions [2,3,4,5]. In addition, the mismatch in Young’s modulus between Ti–6Al–4V and human bone results in stress shielding of the adjacent bone stock, which, in the worst case, can lead to aseptic implant loosening [6]. To inhibit the release of cytotoxic metal ions into the human organism and to improve the mechanical fit between bone and implant, recent developments have focused on low-modulus β-type titanium (Ti) alloys being composed exclusively of biocompatible components [4,7,8,9,10,11,12,13,14].

Biomedical β-type Ti alloys are typically formed in binary system Ti–Nb [7,13,15], ternary systems Ti–Nb–Ta [16,17,18], Ti–Nb–Zr [10], and Ti–Ta–Zr [19], and quaternary systems such as Ti–Nb–Ta–Zr [8,9,11,20] and Ti–Nb–Zr–Sn [21,22]. Entirely β-type Ti alloys may have a Young’s modulus as low as 33 GPa [21], while Ti–6Al–4V has a Young’s modulus of approximately 114 GPa [23]. Niobium and tantalum are particularly promising alloying elements. They stabilize the β-phase in Ti base alloys, form highly corrosion-resistant passive oxide layers, and are biocompatible [13,16,17,24,25,26].

In addition to the intrinsic material properties, processability is a key feature of implant materials. In this context, additive manufacturing (AM) technologies, such as laser beam powder bed fusion (PBF-LB/M), are becoming increasingly important for the fabrication of patient-specific bone implants [27,28,29] with open porous structures [30,31,32,33]. AM allows a high degree of design freedom in the development of new implants [27] but it must be taken into account that the layer-wise manufacturing process is highly anisotropic. Accordingly, additively manufactured parts may exhibit different properties if produced in different build orientations, influencing mechanical performance [15,22,34,35,36,37] and the biological response of human cells [38].

Ti–6Al–4V is by far the best-studied alloy system for application in AM of orthopedic and dental implants, and numerous reviews have been published on this topic [39,40,41,42]. The ultimate tensile strength (UTS) and elongation of additively manufactured Ti–6Al–4V varied in the range of approximately 850–1150 MPa and 3–20% [43], respectively, indicating sufficient strength and mostly also sufficient ductility. In addition, it has been observed that the rough surfaces typically formed in additively manufactured Ti–6Al–4V materials do not interfere with osteogenesis [38,44].

However, a critical issue is the material stiffness, i.e., lack of sufficient elasticity of Ti–6Al–4V, regardless of whether it was manufactured conventionally or additively. To overcome this issue, some of the above-mentioned β-type Ti alloys were recently examined for their suitability for AM. It turned out that the most relevant mechanical properties, such as strength and elasticity, were like those observed for conventionally processed materials [7,11,15,45]. Furthermore, the mechanical behavior of β-type Ti alloys can be tailored to achieve desired properties (e.g., low Young’s modulus) by controlling relevant process parameters such as build direction and scanning speed [15,37]. In addition, PBF-LB/M parts possess surface characteristics that allow direct use as a bone implant, without the need of time- and cost-intensive post-processing strategies [28,38]. Since the surface topography is, among other parameters, influenced by the build orientation [38], understanding of its impact is mandantory.

Our intention was to perform a research study on the AM of highly bio-tolerant Ti base alloys. For this purpose, we chose the titanium–niobium–tantalum (Ti–Nb–Ta) alloy system in the Ti-rich domain, since relatively small changes in the Ti:Nb ratio have a significant effect on the phase formation and accordingly mechanical properties [46,47].

Pre-alloyed Ti–xNb–6Ta AM powders with x = 20, 27, 35 (wt. %) were prepared using the electrode induction melting inert gas atomization (EIGA) process and fully characterized. The powders were consolidated using PBF-LB/M, and the resulting specimens have been previously investigated for microstructure and tensile strength [18]. The present follow-up study aims to further determine the mechanical properties using compression tests, also considering the build orientation of the specimens in the AM process in 0°, 45°, and 90°. Furthermore, the biological response of cultivated human osteoblasts with specimens prepared in 0° and 90° build orientation was investigated. The results are compared to additively manufactured Ti–6Al–4V specimens. Finally, open porous Ti–20Nb–6Ta specimens with a face-centered cubic lattice-structure were additively manufactured to investigate the potential improvement of the bone-implant interaction compared with those printed fully dense.

## 2. Materials and Methods

### 2.1. Specimen Manufacturing by Laser Beam Powder Bed Fusion

Ti–Nb–Ta spherical powders with nominal chemical compositions of Ti–20Nb–6Ta, Ti–27Nb–6Ta, and Ti–35Nb–6Ta wt. % were produced by the EIGA process, which was conducted under purified argon (4.6, Linde GmbH, Pullach, Germany) atmosphere from pre-alloyed electrodes (TANIOBIS GmbH, Goslar, Germany). For details on the experimentally determined chemical composition, we refer to Johannsen et al. [18]. 

The raw Ti–Nb–Ta powder materials were sieved through 150 µm meshes, and the remaining powders were transferred into an air classifier for deagglomeration and the removal of fine particles < 10 µm to improve their handling, i.e., to avoid dusting during the manufacturing process and to improve flow properties. The remaining powder was sieved with ultrasonic vibration through 63 µm stainless steel mesh.

Particle size distributions (PSD) determined using a Master Sizer 2000 (Malvern, Worcestershire, UK) indicated that there is no evidence of particles <15 µm. D95 was between 63 µm and 69 µm for the different powders. In addition, a commercially available titanium grade 23 powder (Ti64-53/20, Tekna Plasma Europe, Macon, France) was used for manufacturing the reference Ti–6Al–4V specimens.

The Ti–Nb–Ta specimens were produced using a DMP350 Flex (3D Systems Corp., Rock Hill, SC, USA) equipped with a 1 kW single-mode laser (YLR-1000-WC-Y14, IPG Laser GmbH, Burbach, Germany), whereas Ti–6Al–4V was processed using a SLM500 Quad (Nikon SLM Solutions AG, Lübeck, Germany). Argon gas was used during fabrication to prevent oxidation. A stripe-based scanning strategy [18] was used with hatch distances of 69 µm for the Ti–Nb–Ta alloys and 100 µm for Ti–6Al–4V.

The process parameters for the AM of all Ti–Nb–Ta alloys were adopted from Johannsen et al. (Table 1) [18], providing specimens with densities >99.96% and microstructures with a homogeneous element distribution. Process parameters for AM of Ti–6Al–4V specimens were taken from an IAPT (Research Institution for Additive Manufacturing Technologies IAPT, Hamburg, Germany) database (Table 1). In addition, Ti–6Al–4V specimens were subjected to a stress relief annealing under vacuum (800 °C, 2 h). Such heat treatment was not required for the additively manufactured Ti–Nb–Ta specimens and was, therefore, not performed.

#### 2.1.1. Specimens for Mechanical Characterization

Cylindrical specimens were fabricated at 0°, 45°, and 90° (90° corresponds to parallel orientation to the direction of construction) for compression testing of all dense Ti–Nb–Ta alloys using the process parameters described above. Accordingly, nine different groups were investigated: [Ti–20Nb–6Ta, Ti–27Nb–6Ta, Ti–35Nb–6Ta] × [0°, 45°, 90°] with n = 5 specimens each.

Similar to a report by Schulze et al. [7], their dimensions were d × l = 10.7 mm × 16.0 mm in the as-printed state. The specimens were milled to d × l = 6.9 mm × 10.4 mm (Figure 1A). A black and white speckle pattern was applied to the surface of the specimens for local strain measurement using digital image correlation (DIC).

Open porous lattice-structured specimens were designed in Creo Parametrics 6.0.3.0 (PTC Inc., Boston, MA, USA) using a 1 × 1 × 1 mm^3^ face-centered cubic unit cell with longitudinal struts comparable to Li et al. [30]. The strut diameter was 0.3 mm, and the resulting pore size was 0.6 mm, leading to a porosity of 70%. These values were designed based on previous studies where the defined pore size and porosity were reported to provide good osseointegration [33]. For the compression tests, n = 5 rectangular specimens (7 × 7 × 11 mm^3^) of Ti–20Nb–6Ta and Ti–6Al–4V were manufactured by PBF-LB/M. The rectangular specimens and the designed unit cell are shown in Figure 1B.

µCT scans with a resolution of 9 µm were obtained (Skyscan1076, Bruker, Billerica, MA, USA) to define the porosity of the manufactured open porous specimens for mechanical characterization. The voltage and current were set at 95 kV and 104 μA, and a 0.5 mm aluminum filter was used. The porosity of the specimens was measured using Materialise Mimics 25.0 (Materialise NV, Leuven, Belgium). In addition, strut thickness and pore size were measured using a digital microscope (VHX-6000, Keyence Corporation, Osaka, Japan) at 10 different locations (5 on frontal and 5 on top view) on each specimen. Since the manufacturing of the Ti–20Nb–6Ta specimens resulted in an increase in strut thickness and corresponding decrease in pore size and total porosity, the Ti–6Al–4V specimens had to be adjusted (strut diameter of 0.39 mm) to have an almost similar geometry for comparability of the results. The measured geometric parameters are shown in Table 2.

#### 2.1.2. Specimens for Biological Characterization

The biological characterization was performed with both densely and open porous lattice–structured specimens. Cylindrical dense specimens (d = 12 mm, h = 2 mm) were fabricated in 0° and 90° build orientation, leading to six different groups [Ti–20Nb–6Ta, Ti–27Nb–6Ta, Ti–35Nb–6Ta] × [0°, 90°] with n = 38 specimens each. Additionally, specimens made of Ti–6Al–4V serving as a reference were manufactured in 0° and 90° build orientations. The surfaces were not further processed and are referred to as “as-printed” in the following. Open porous lattice-structured specimens were manufactured consisting of one layer of the above-described face-centered cubic unit cells (Figure 1D). Accordingly, the specimens were 12 mm in diameter and 2 mm in height, i.e., 1 mm of dense structure and 1 mm of open porous lattice structure. Prior to their biological testing, all specimens were cleaned in an ultrasonic bath to remove residual powder and heat-sterilized at 180 °C for 135 min.

The surface roughness of the dense specimens was analyzed with a laser-scanning microscope (VK-X250, Keyence Germany GmbH, Neu-Isenburg, Germany) with 20 times magnification, λ_S_ = 8, and λ_C_ = 25. The roughness values are summarized in Table 3, and light microscopic images are shown in Figure 1E. The light microscopic images demonstrated melt tracks and partly melted particles on the surface.

### 2.2. Microstructural Characterization

In a previous study, the microstructure of the Ti–Nb–Ta alloys was described in detail [18]. Here we give a brief summary of the main findings, as they are crucial for understanding the present observations. Furthermore, the microstructure as a function of the orientation in the build chamber was studied by backscattered electron analysis (BSA). In addition, energy-dispersive X-ray spectroscopy (EDX) images of Ti–27Nb–6Ta in 0° and 90° orientation are shown in Figure A1 to illustrate the chemical homogeneity.

### 2.3. Mechanical Characterization

The cylindrical specimens were compressively loaded with 0.005 mm/s in a universal testing machine (100 kN Landmark^®^, MTS Systems Corporation, Eden Prairie, MN, USA). The test was terminated when the specimen failed (drop in the force–displacement curve) or when a compressive stress of 1900 MPa was reached. Furthermore, the elongation in load direction was measured using a 2D-DIC system (camera: isi-sys GmbH, Kassel, Germany; image recording: ViC Snap, Correlated Solutions Inc, Irmo, SC, USA), and the data were analyzed in GOM Suite 2021 (Carl Zeiss GOM Metrology GmbH, Braunschweig, Germany). The DIC-based elongation was used to calculate the compressive modulus (C in GPa). In addition, the compressive yield strength at 0.2% plastic strain (σ_C,0.2_ in MPa), the ultimate compression strength (UCS), plastic elongation at break (ε_B_ in %), and the elastic admissible strain (EAS) were evaluated. The EAS is defined as an indicator of mechanical biofunctionality and was calculated as the ratio of the yield strength to the modulus [10,15]. This parameter has been used previously and is recommended for assessing the suitability of an implant material from a mechanical point of view [10,15]. Representative areas of the fracture surfaces were analyzed by a field emission scanning electron microscope (SEM, MERLIN^®^ VP Compact, Co. Zeiss, Oberkochen, Germany). The specimens were mounted on an Al-SEM carrier with adhesive conductive carbon tape (Plano GmbH, Wetzlar, Germany).

The open porous lattice-structured specimens were compressively loaded until failure using a universal testing machine (Z050-50kN, Zwick Roell, Ulm, Germany) with a crosshead speed of 0.005 mm/s. Force–displacement curves were obtained using a tactile extensometer (digiClip Extensometer, Zwick Roell, Ulm, Germany), and mechanical properties were evaluated similar to those of densely manufactured specimens. For the calculation, the nominal cross-section of 7 × 7 mm^2^ was used. The apparent stiffness of the open porous lattice-structured specimens is further designated as compressive modulus.

### 2.4. Biological Characterization

#### 2.4.1. Cell Biological Experiments

The biological characterization of dense and open porous lattice-structured specimens was performed with eight independent human osteoblast cell cultures (female: n = 4, mean age: 77 ± 3.7 years; male: n = 4, mean age: 78 ± 2.4 years). The local ethics committee approved the collection of cells after written informed consent of the patients (ethical approval of University of Rostock from 12/06/2018, registration no.: A2010-0010). The protocols for cell isolation from the cancellous bone of femoral heads of patients undergoing primary hip replacement and subsequent cultivation were previously reported [48]. Human osteoblasts from passage 4 were used for experiments. For this purpose, the cells were cultivated under standard cell culture conditions in a humified atmosphere of 37 °C and 5% CO_2_ in calcium-free Dulbecco’s Modified Eagle Medium (DMEM) containing 10% fetal calf serum (FCS; both: PAN-Biotech, Aidenbach, Germany), 1% amphotericin B, 1% penicillin–streptomycin, and 1% HEPES buffer (all: Sigma-Aldrich, Munich, Germany). To maintain the osteogenic phenotype, 10 mM β-glycerophosphate, 50 μg × mL^−1^ ascorbic acid, and 100 nM dexamethasone were added to the cell culture medium (all: Sigma-Aldrich, Munich, Germany). Each cell number of 250,000 cells per 50 µL cell culture medium was seeded on top of the cleaned and sterilized specimens, allowing initial adherence over 30 min. Afterward, the cell-seeded specimens were incubated with 1 mL cell culture medium with additives as described before. To enhance the mineralization capacity of the human osteoblasts, the cell culture medium was supplemented with calcium chloride dehydrate (CaCl_2_ * 2 × H_2_O, final concentration: 1.8 mmol × L^−1^). The cultivation of osteoblasts on the dense and open porous specimens was conducted over 7 and 14 days. In the latter case, the medium was changed after 7 days.

To analyze the cell spreading after 7 days on the surface of densely manufactured specimens, the samples were washed with phosphate-buffered saline (PBS; Biochrom AG, Berlin, Germany), and cells were fixed with fixation buffer (1% paraformaldehyde, 2.5% glutaraldehyde, 0.1 M sodium phosphate buffer, pH 7.3) and stored at 4 °C. Fixed samples were washed with sodium phosphate buffer (0.1 M) and subsequently dehydrated in an ascending series of ethanol prior critical point drying using CO_2_ as an intermedium (Emitech K850, Quorum Technologies LTD, East Sussex, UK). The specimens were sputter coated with a thin layer of gold with a thickness of approximately 15 nm under vacuum in an argon atmosphere (SCD 500 Leica, Wetzlar Germany). Using a field emission scanning electron microscope (MERLIN VP Compact, Carl Zeiss, Oberkochen Germany), the surface spreading of osteoblasts was imaged from selected regions (applied detector: HE-SE2; accelerating voltage 10.0 kV and 5.0 kV, working distance 3.9 mm).

#### 2.4.2. Analysis of Cell Mediators Involved in Bone Formation, Remodeling, and Inflammation

The biological response of human osteoblasts on the specimens was characterized after 7 and 14 days of cultivation. Different mediators involved in osteogenesis, bone remodeling, and inflammation were analyzed by quantifying specific mRNA transcripts or by quantifying secreted proteins in the supernatants.

For gene expression analyses, the RNA of cells seeded on top of the specimens was isolated with the innuPREP RNA Mini Kit 2.0 (Analytik Jena, Jena, Germany) according to the manufacturer’s instructions. Only for cell lysis was the protocol modified with regard to a more prolonged incubation with the lysis buffer. Here, the osteoblasts have been lysed for 15 min under constant shaking to enhance the quality of isolated RNA. Before RNA was transcribed into cDNA using the High-Capacity cDNA Reverse Transcription Kit (Applied Biosystems, Foster City, CA, USA) according to the manufacturer’s instructions, its concentration was measured with a microplate reader and the NanoQuant^TM^ Plate (both: Tecan Trading AG, Maennedorf, Switzerland). Afterward, 50 ng (for the dense specimens) or 100 ng (for the open porous specimens) RNA was used for the reverse transcription protocol (total volume of RNA and Mastermix was 20 µL). The PCR was performed using the following protocol: 10 min at 25 °C, 120 min at 37 °C, and 5 min at 85 °C in a thermocycler (Analytik Jena, Jena, Germany). Finally, all samples were diluted with 20 µL of RNase-free water and stored at −20 °C until further use. 

Relative quantification of gene expression of defined genes was determined with the innuMIX qPCR DSGreen Standard in a qTower 2.0 (both: Analytik Jena AG, Jena, Germany). All primers used for analysis are listed in Table 4.

For each gene of interest, a master mix with the respective forward and reverse primer at 0.5 µL each, 3 µL of distilled water, and 5 µL of innuMIX qPCR DSGreen was prepared. Afterward, 1 µL of template cDNA was pipetted onto the bottom of a 96-well PCR plate in duplicates, and a volume of 9 µL of the master mix was added. RNase-free water served as a negative control. The PCR plate was sealed, and qPCR followed the protocol: 2 min at 95 °C, 40 cycles of 5 s at 95 °C, and 25 s at 60 °C. A cycle threshold (Ct) of 30 was set as the interpretation limit. The ∆∆Ct method was performed using the Equation (1), where ∆∆Ct values are depicted as 2−∆∆Ct.
(1)$$∆∆Ct={∆Ct}_{Ti-xNb-6Ta}-{∆Ct}_{Ti-6Al-4V}$$

The amount of secreted proteins in the supernatants was determined via enzyme-linked immunosorbent assays. Specifically, the protein levels of cross-linked C-telopeptides of type I collagen (CICP, MicroVue Quidel, San Diego, CA, USA), interleukin (IL-) 6, and IL-8 (Thermo Fisher Scientific Inc., Waltham, MA, USA) were quantified according to the manufacturer’s instructions. Absorbance was measured at 405 nm (CICP) or 450 nm (IL-6, IL-8) using a microplate reader (Tecan Trading AG, Maennedorf, Switzerland). Defined standard curves were used to calculate the protein concentrations, respectively. Finally, the protein concentration of the samples was related to the total protein content, which was determined with the Qubit Protein Assay Kit and Qubit 1.0 (both: Invitrogen, Waltham, MA, USA).

### 2.5. Statistical Evaluation

The statistical analysis and graphical illustration were performed in GraphPad Prism 9.2 (GraphPad Software, San Diego, CA, USA). Unless otherwise noted, data are presented as single values with median and interquartile range (figures) or mean and standard deviation (tables). Compression test results of n = 5 specimens for each group were tested for significance using the Mann–Whitney U test. Pairwise comparisons were made for each alloy according to build orientation and for each build orientation according to niobium content (e.g., Ti–20Nb–6Ta 0° vs. Ti–20Nb–6Ta 45° and vs. Ti–20Nb–6Ta 90°; Ti–20Nb–6Ta 0° vs. Ti–27Nb–6Ta 0° and vs. Ti–35Nb–6Ta 0°). For the biological analyses, a minimum of four different osteoblastic donors was used, and comparisons between experimental groups were performed using a paired or unpaired *t*-test. The level of significance was *p* < 0.05 for all tests.

## 3. Results

### 3.1. Microstructure Characterization

A detailed study on the chemical composition (Energy-dispersive X-ray spectroscopy, EDS) and phase composition (X-Ray diffraction analysis) of Ti–20Nb–6Ta, Ti–27Nb–6Ta, and Ti–35Nb–6Ta has already been published in [18]. All three Ti–Nb–Ta alloys possess a homogenous element distribution, which was also observed in specimens of this present study (see Appendix A for an example). In line with pure Nb, Ti–35Nb–6Ta revealed reflections attributed to the bcc β-phase. Increasing the Ti content leads to significant changes in the phase constitution and the appearance of more complicated diffraction patterns. While the reflections in the diffractogram of Ti–27Nb–6Ta could not be assigned unequivocally to specific phases, those of Ti–20Nb–6Ta clearly indicated the formation of an orthorhombic α″-phase. It was supposed that, accordingly, the microstructure of Ti–27Nb–6Ta contains both α″- and β-phase shares. A comparison of the microstructure using light microscopy visualized another difference in Ti–20Nb–6Ta on the one hand and Ti–27Nb–6Ta as well as Ti–35Nb–6Ta on the other hand. While the latter displayed typical features of additively manufactured specimens, c.f. melt pool boundaries and elongated columnar grains in build direction, the light microscopic images of Ti-20Nb-6Ta additionally exhibited platelet-like structures. However, those could not be differentiated by means of energy-dispersive X-ray spectroscopy.

The BSA images of Ti–20Nb–6Ta, Ti–27Nb–6Ta, and Ti–35Nb–6Ta additively built in 0° and 90° direction shown in Figure 2 confirm the previous observations from the light microscopic investigations. The chemical, i.e., phase composition, has a significant influence on the morphology of the additively manufactured samples. Regardless of the chemical composition, melt pool boundaries, indicated by white triangles, are clearly visible in both 0° and 90° build orientations. In addition, the 90° images show elongated columnar grains oriented along the build orientation (grain boundaries are indicated by white dotted lines), reflecting the high anisotropy of the AM process due to epitaxial solidification. In contrast to Ti–27Nb–6Ta and Ti–35Nb–6Ta, the microstructure of Ti–20Nb–6Ta additionally shows plate-like segregations, which is fully consistent with the observations made by light microscopy [18]. Their size is difficult to determine from the available BSE images, but their thickness can be estimated to be 1–2 µm.

### 3.2. Mechanical Characterization by Compression Testing

Stress–strain curves of one dense and one open porous specimen, representative of each group, are shown in Figure 3A. Specimens containing 20 wt. % and 27 wt. % of niobium fractured at ~45° to the load axis. Ti–35Nb–6Ta specimens did not fracture up to 1900 MPa compressive stress applied. Dimples and cleavage facets were observed on the fracture surfaces of Ti–20Nb–6Ta (Figure 3B,C), indicating the mixed behavior of ductile failure and brittle fracture, respectively. Only the fracture surfaces of the 0° and 45° Ti–20Nb–6Ta specimens could be examined, as the fractured parts of the other specimens adhered to each other, although the macroscopic fracture was visible. The open porous lattice-structured specimens also fracture at ~45° to the load axis, and the fractured pieces adhered to each other. Open porous Ti–20Nb–6Ta specimens showed ductile behavior with a pronounced plateau after yielding. In contrast, open porous Ti–6Al–4V specimens were less ductile with stepwise fracture of the lattice structure (Figure 3A).

Both the alloy composition, i.e., Ti:Nb ratio and the build orientation, affected the mechanical properties of the Ti–Nb–Ta alloys. The experimentally determined mechanical properties are graphically presented in Figure 4 and exact values of the mechanical properties, including mean and standard deviation and calculated elastic admissible strains, are given in Table 5.

For each alloy, the compressive modulus and yield strength as a function of the build orientation can be ordered as follows: 90° < 0° < 45°. Accordingly, the Ti–20Nb–6Ta in 90° build orientation showed the highest elastic admissible strain. The ultimate compressive strength and elongation at the breaks of Ti–20Nb–6Ta and Ti–27Nb–6Ta were minimal in 45° build orientation, and parameters increased with the niobium content. Also, for these two alloys, specimens manufactured in 45° build orientation showed the lowest strength and elongation at break, whereas small differences were visible between 0° and 90° build orientation. Ti–35Nb–6Ta showed a dissimilar behavior, the highest yield strength in 45° build orientation, and no fracture in all build orientations until a maximum compressive stress of 1900 MPa.

The compressive modulus of open porous lattice-structured Ti–20Nb–6Ta specimens compared to dense Ti–20Nb–6Ta specimens was reduced from 61.7 ± 12.9 GPa (averaged over different build orientations) to 8.7 ± 1.0 GPa. Open porous Ti–20Nb–6Ta specimens showed significantly lower compressive modulus and strength values but increased ductility compared to similar Ti–6Al–4V specimens.

### 3.3. Biological Characterization 

Within the biological characterization, the influence of the chemical composition, build orientation during PBF-LB/M, and cultivation time was considered.

Regarding the influence of the build orientation of the Ti–Nb–Ta specimens on cell growth, it is evident that the cell settlement depends on it (Figure 5A). Scanning electron microscopy showed that the surfaces manufactured perpendicular to the building direction (0°) appeared smoother. Here, the osteoblasts, when examined in areas of high cell counts, had an elongated and spindle-shaped appearance and were located close together in clusters. This growth behavior of the cells resulted in a low degree of cell spreading and incomplete coverage of the surface. In contrast, cells exhibited a flat, stretched-out morphology on the surfaces of specimens manufactured in 90° build orientation, covering most of the material surface area. The surface morphology of specimens manufactured in 0° and 90° build orientation showed differences, i.e., 90° specimens showed a higher amount of partly melted particles on the surface. These were firmly attached to the surface and were caused by incomplete melting during the PBF-LB/M process. These elevations were clearly surrounded by the cells. Comparing the respective alloys with each other, it is noticeable that cell spreading on Ti–20Nb–6Ta was more pronounced than on Ti–27Nb–6Ta and Ti–35Nb–6Ta. Finally, osteoblasts seeded on Ti–6Al–4V samples for comparison showed a large number of round cells for both build orientations, with the 90° build orientation having a greater influence.

In addition to the cell spreading on the surface of the specimens, the build orientation and the cultivation time significantly affected the gene expression profile of osteogenic differentiation markers. For *ALP*, *BGLAP*, and *SPP1*, differences to the Ti–6Al–4V control were apparent, but no statistical differences were shown by the build orientation or cultivation time due to a high donor variability (Figure 5B–D). *RUNX2* mRNA was induced on the Ti–Nb–Ta specimens after 7 d of cultivation without differences between build orientations (Figure 5E,F). However, after 14 days, a trend of reduced *RUNX2* mRNA was more present in osteoblasts cultivated on 0° built orientation specimens. Regarding the gene expression influenced by the niobium content, a significant induction of *RUNX2* was observed for Ti–20Nb–6Ta after 7 d (*p* = 0.0411 compared to Ti–6Al–4V) and for Ti–35Nb–6Ta after 14 d (*p* = 0.0158 compared to Ti–6Al–4V). The gene expression of *COL1A1* was significantly influenced by the build orientation and cultivation time. After 7 d of cultivation, compared to osteoblasts on Ti–6Al–4V references, a significant increase in *COL1A1* mRNA was determined for Ti–20Nb–6Ta with 0° build orientation (*p* = 0.0118) and for all Ti–xNb–6Ta specimens with 90° build orientation (*p* = 0.0019 [Ti–20Nb–6Ta], *p* = 0.0021 [Ti–27Nb–6Ta], *p* = 0.0011 [Ti–35Nb–6Ta]). In osteoblasts cultivated on Ti–20Nb–6Ta (*p* = 0.0350) and Ti–35Nb–6Ta (*p* = 0.0133), a significantly higher gene expression rate in 90° was present compared to 0° build orientation. After 14 d, the *COL1A1* mRNA was on the same transcript level as those for Ti–6Al–4V. This reduction was significantly different for osteoblasts cultivated on Ti–20Nb–6Ta (*p* = 0.0129) and Ti–35Nb–6Ta (*p* = 0.0148) specimens (Figure 5G,H). The release of the cross-linked C-telopeptides of Type 1 collagen (CICP) was reduced in osteoblasts cultivated on Ti–xNb–6Ta after 7 d compared to those on Ti–6Al–4V (Figure 5I). This reduction was significant for osteoblasts on Ti–27Nb–6Ta 0° build orientation (*p* = 0.0110). After 14 days, a decreased release of CICP was apparent for osteoblasts cultivated on 0° specimens. In contrast, on 90° samples, CICP synthesis rates remained stable except for osteoblasts on Ti–6Al–4V with significantly reduced CICP protein levels (*p* = 0.0126 compared to 7 d, Figure 5J). Significantly higher CICP protein was determined for osteoblasts cultivated on Ti–27Nb–6Ta specimens with 90° compared to 0° build orientation (*p* = 0.0477).

Gene expression analysis was further used to determine the bone-destructive metalloprotease *MMP1* and its natural inhibitor *TIMP1*. The induction of *MMP1* mRNA after 7 days was particularly influenced by the build orientation (Figure 6A). While in specimens with 0° orientation, increased values were found for osteoblasts cultivated on Ti–20Nb–6Ta and Ti–27Nb–6Ta, cells on 90°-oriented samples showed significantly reduced gene expression (*p* = 0.0184 [Ti–20Nb–6Ta compared to Ti–6Al–4V], *p* = 0.0250 [Ti–27Nb–6Ta compared to Ti–6Al–4V]). A significant difference between the build orientations was only detected for Ti–20Nb–6Ta (*p* = 0.0239). Furthermore, *MMP1* was not affected on Ti–35Nb–6Ta test samples in 0° orientation, whereas a non-significant reduction was detected on 90° specimens (Figure 6A). After 14 days, a similar trend was observed to that observed after 7 days (Figure 6B). *MMP1* gene expression was significantly increased on Ti–20Nb–6Ta (*p* = 0.0018) specimens with 0° orientation compared to Ti–6Al–4V, while a significant reduction was evident at 90° orientation (*p* = 0.0422). In addition, the build orientation significantly affected cells cultivated on Ti–20Nb–6Ta (*p* = 0.0124). Moreover, a significant difference in niobium content was also observed after 14 days: osteoblasts on Ti–27Nb–6Ta at 0° orientation showed significantly less *MMP1* mRNA than Ti–20Nb–6Ta (*p* = 0.0332). In contrast to *MMP1*, the gene expression of *TIMP1* was not strongly affected (Figure 6C,D): after 7 days of cultivation, *TIMP1* mRNA of cells was unchanged on all Ti–Nb–Ta alloys with 0° orientation compared to cells on Ti–6Al–4V. On 90°-oriented surfaces, there was a trend toward decreased gene expression levels depending on the niobium content, but no significant differences could be determined (Figure 6C). A similar gene expression profile was also detected after 14 days, with *TIMP1* mRNA significantly reduced in osteoblasts on Ti–27Nb–6Ta with 0° orientation (*p* = 0.0312) and on Ti–20Nb–6Ta with 90° orientation (*p* = 0.0315; Figure 6D).

Further, the release of interleukin (IL- 6 and IL-8) was determined to evaluate the pro-inflammatory potential of Ti–Nb–Ta specimens. For IL-6, significantly higher protein levels (*p* = 0.0022 [Ti–20Nb–6Ta], *p* = 0.0343 [Ti–27Nb–6Ta], *p* = 0.0005 [Ti–35Nb–6Ta] compared to 90°) were determined for osteoblasts cultivated on specimens in 0° build orientation after 7 d. Moreover, a significantly lower protein synthesis rate than those of osteoblasts on Ti–6Al–4V was apparent for Ti–27Nb–6Ta (*p* = 0.0072) and Ti–35Nb–6Ta (*p* = 0.0207). Osteoblasts on Ti–20Nb–6Ta released significantly less IL-6 than cells on Ti–35Nb–6Ta (*p* = 0.0367). Additionally, cells on 90°-oriented Ti–Nb–Ta specimens secreted IL-6 to a lesser extent than osteoblasts on Ti–6Al–4V (Figure 7A). After 14 d, IL-6 secretion remained constant for osteoblasts on 0° Ti–Nb–Ta specimens. However, the IL-6 release of cells on Ti–35Nb–Ta was higher than those on Ti–6Al–4V (*p* = 0.0207). Although the release of IL-6 decreased after 14 days for osteoblasts on 90° Ti–6Al–4V, there was a higher synthesis rate for cells on 90°-oriented Ti–Nb–Ta specimens, with significant differences for osteoblasts on Ti–20Nb–6Ta (*p* = 0.0212) and Ti–27Nb–6Ta (*p* = 0.010, Figure 7B). For IL-8, significantly higher protein synthesis rates on day 7 were present for osteoblasts on specimens with 0° build orientation (*p* = 0.0094 [Ti–20Nb–6Ta], *p* = 0.0274 [Ti–27Nb–6Ta], *p* = 0.0076 [Ti–35Nb–6Ta]; compared to 90°). On samples with 90° build orientation, significantly reduced IL-8 was determined for osteoblasts on Ti–Nb–Ta compared to those on Ti–6Al–4V (*p* = 0.0424 [Ti–20Nb–6Ta], *p* = 0.0073 [Ti–27Nb–6Ta], *p* = 0.0301 [Ti–35Nb–6Ta]; Figure 7C). On day 14, constant IL-8 protein was detectable for osteoblasts on 0° oriented test specimens (Figure 7D). Although there was a significant increase in IL-8 protein for osteoblasts on 90° oriented Ti–Nb–Ta specimens (*p* = 0.0250 [Ti–20Nb–6Ta], *p* = 0.0128 [Ti–27Nb–6Ta], *p* = 0.0104 [Ti–35Nb–6Ta]; compared to 7 d), the amount of IL-8 was decreased compared to cells cultivated on Ti–6Al–4V (Ti–35Nb–6Ta: *p* = 0.0448).

The biological characterization of open porous Ti–20Nb–6Ta specimens was performed with regard to the differentiation capacity, induction of inflammation, and bone remodeling of human osteoblasts. It was compared to similar specimens made of Ti–6Al–4V. A time-dependent induction of the osteoblastic differentiation was apparent in osteoblasts on Ti–20Nb–6Ta. In detail, significantly enhanced *RUNX2* transcripts were determined after 14 d compared to osteoblasts seeded on Ti–6Al–4V (*p* = 0.0415, Figure 8A). For *COL1A1*, an increased gene expression was observed in both time intervals (*p* = 0.0314 [day 7, compared to Ti–6Al–4V]) with higher but non-significant values after 14 days. Regarding the induction of bone remodeling, mRNA transcript levels of the collagenase *MMP1* and its natural inhibitor *TIMP1* were determined, indicating decreased *MMP1* transcripts (*p* = 0.0418 [day 14, compared to Ti–6Al–4V] and a time-dependent increase of *TIMP1* gene expression. Moreover, a significant decrease in IL-6 gene expression after 14 days (*p* = 0.0327, compared to 7 days) was observed in osteoblasts cultivated on open porous Ti–20Nb–6Ta specimens. 

In addition to gene expression analyses, the release of CICP and IL-6 in cell culture supernatant was also determined. No differences in the amount of CICP from osteoblasts on Ti–20Nb–6Ta and Ti–6Al–4V could be detected. A time-dependent decrease in protein was detected for both open porous specimens, although this was significant only for osteoblasts on Ti–6Al–4V (*p* = 0.0148, Figure 8B). A similar trend was also found for the secretion of IL-6, with a significant decrease in the amount of protein for Ti–20Nb–6Ta (*p* = 0.0009, Figure 8C).

## 4. Discussion

In this study, we characterized the mechanical and biological properties of Ti–xNb–6Ta (x = 20, 27, 35) alloys manufactured by PBF-LB/M. The influence of build orientation during manufacturing on different properties was evaluated. In addition to fully dense specimens, open porous Ti–20Nb–6Ta specimens with a defined lattice structure were also characterized and compared to Ti–6Al–4V specimens.

The compressive modulus of the Ti–Nb–Ta alloys ranged from 43.7 ± 2.7 GPa (Ti–20Nb–6Ta, 90°) to 85.9 ± 2.7 GPa (Ti–27Nb–6Ta, 45°). Even though this is significantly higher than that of human cortical (18.6 GPa [49]) or cancellous bone (0.2–4.5 GPa [50]), it is distinctly reduced compared to Ti–6Al–4V (Young’s modulus: ~114 GPa [22]). In addition to the compressive modulus, the elastic admissible strain, which is defined as the ratio of yield strength to elasticity, is a useful parameter for assessing the potential of an implant material [10]. Elastic admissible strains of, e.g., Ti–42Nb are 0.95–1.54% [7,15], of various Ti–Nb–Zr alloys 1.08–1.31% [10], of various Ti–Nb–Ta–Zr alloys 0.71–1.27% [9], and of Ti–6Al–4V ~0.8% [23]. Ti–Nb–Ta alloys investigated in this study showed elastic admissible strains between 0.96% and 1.44%, thus indicating a high ratio of compressive modulus and yield strength. Specimens built in 90° orientation showed the highest admissible strain for each alloy and are ranked as follows: Ti–20Nb–6Ta > Ti–27Nb–6Ta > Ti–35Nb–6Ta.

Both the niobium content and the build orientation of the Ti–Nb–Ta specimens affected the compressive mechanical properties. As reported by Johannsen et al. [18], the Ti:Nb ratio in Ti–Nb–Ta alloys influences the microstructure and the phase composition of the powders and the additively manufactured parts. It was reported that Ti–20Nb–6Ta is mostly composed of an orthorhombic α″ martensite phase whereas Ti–35Nb–6Ta crystallizes entirely in a bcc β-phase. Ti–27Nb–6Ta is characterized by a structure in which probably ω-, α″-, and/or β-phases coexist.

Since the specimens in the present study were manufactured using the same powder feedstock and the AM process parameters developed in [18], it is obvious that identical microstructures were obtained compared to those published. This can be concluded from Figure 2 in which the microstructures of three Ti–Nb–Ta samples are compared by BSE analysis. Independent of the orientation in the built chamber, the BSE analysis of Ti–20Nb–6Ta shows platelet-like features, whereas Ti–27Nb–6Ta and Ti–35Nb–6Ta do not. This is in full agreement with the results of the previous light microscopic investigations in [18]. 

The different microstructures of the additively manufactured specimens cause different fracture behavior. Ti–35Nb–6Ta with a bcc structure did not fracture under compressive load. This is a behavior frequently observed for entirely β-phase Ti alloys such as Ti–42Nb [7] or Ti–Nb–Zr (Nb + Zr > 59 wt. %) [10]. For Ti–Nb–Ta–Zr alloys with a mixed β- and ω-phase, fracture at 45° to the load axis and the adhesion of the fractured pieces have been reported [8], identical to our observations. The compressive fracture of Ti–6Al–4V–5Nb consisting of an α′–β mixed phase with appearance of dimples and cleavage facets was reported by Sui et al. [51].

In addition to the chemical composition, the build orientation and scanning strategy during PBF-LB/M of β-Ti alloys influences the microstructure and, thus, the mechanical properties [15,37,52]. It is well understood that during fabrication, elongated columnar β-grains may form along the build direction [15,52] and that their formation is strongly influenced by the scanning strategy and the depths of the melt pools [37,52]. In line with the observations of Pilz et al. [15], Young’s modulus (in our case compressive modulus) can be ordered as follows from highest to lowest: 45° > 0° > 90° (build orientation). In AM, the epitaxial growth of crystals is influenced by the movement of melt pools, which in turn depends on spatial and temporal variations in the temperature gradient. This can be used to program a preferred crystallographic orientation to feature desired properties [37]. The lowest stiffness of an additive manufactured part is achieved by parallel alignment of the <001> crystal direction to the load direction [15]. The results of our study indicate the possibility of the Ti–Nb–Ta alloys to be used as material for bone implant applications and that the stiffness, which is crucial in terms of implant-to-bone load transfer, can be tailored by the build orientation.

Although the findings of the present study are in general agreement with those of Pilz et al. [15], they did not observe significant differences in strength and ductility in specimens manufactured with different build orientations. The ductility of conventionally manufactured metal specimens can be traced back to grain sliding. In additively manufactured specimens, this effect can be severely limited due to the smaller crystallite sizes, the occurrence of melt pool boundaries and the cellular structure [36,53]. Liu et al. [53] investigated the deformation mechanism of a high-entropy alloy, Ti–Nb–Ta–Zr–Mo, produced by additive manufacturing. Their results suggested that the strength and ductility of the alloy resulted from a complex microstructure that included dislocations, solidification cell patterns, and grain boundaries. The study highlighted that the occurrence of dislocation slip, either within a single cell or across multiple cells, played a critical role. Despite the homogeneous element distribution of the Ti–Nb–Ta alloys, they have unique 1–2 µm thick melt pool boundaries with a slight chemical segregation of Ti- and Nb/Ta-rich zones and platelet-like structures in Ti–20Nb–6Ta [18]. In the context of the complex deformation mechanisms of other β-type Ti alloys, we relate the dependence of strength and ductility on build orientation and chemical composition to the microstructure. However, a detailed investigation of the deformation behavior comparable to that of Liu et al. [53] was beyond the scope of our present study, but should be addressed in future research.

To further reduce the mechanical mismatch in elasticity between human bone and implant material and to improve the bone-implant interaction, open porous lattice-structured Ti–20Nb–6Ta specimens with a face-centered unit cell were additively manufactured (Figure 1). They have a compressive stiffness of 8.7 ± 1.0 GPa, which is in the range of human bone. Even though this is higher than the stiffness of porous structures reported in other studies [30,32], the observed yield (179.2 ± 4.5 MPa) and ultimate compressive strength (351.7 ± 7.5 MPa) are higher than in these studies and higher than the strength of human bone [54]. Compared to porous Ti–6Al–4V specimens with similar structure, Ti–20Nb–6Ta possesses a significantly lower compressive modulus but also lower strength, most probably due to the different intrinsic material properties. Nevertheless, it can be concluded that AM of open porous structured Ti–20Nb–6Ta is a suitable approach for the physical functionalization of orthopedic implants to enhance the bone–implant interaction.

Commonly used Ti–6Al–4V implants have significant disadvantages with regard to the toxicity of Al and V, when dissolved in the human organism [2,3,4,5]. In contrast, binary or ternary alloys with Ti, Nb and /or Ta as alloying constituents exhibit excellent biocompatibility and high corrosion resistance [55]. Last but not least, tantalum, with its low toxicity and high corrosion resistance, is an ideal candidate for bone regeneration [24,56]. In previous studies, we confirmed that both Ti–Nb alloys possess good biocompatibility [57] and that tantalum scaffolds are excellent substrates for bone cells depositing [58]. Accordingly, Ti–Nb–Ta alloys appear to be materials for the application as bone substitutes, promoting osteogenesis.

The surfaces of PBF-LB/M processed specimens typically have considerable roughness due to incompletely melted particles [28], and the surface properties are influenced by, among other parameters, the build orientation [38]. However, Ginestra et al. [38] reported a limited influence of the build orientation (0°,15°, 30°, 45°) on the long-term osseointegrative behavior, with roughnesses (Ra) ranging from 17.4 ± 1.8 µm to 24.4 ± 3.1 µm. In line with previous studies [38,44], we demonstrated that as-printed surfaces do not impair osteogenesis and, therefore, no post-processing is necessarily required. In contrast to the observations of Ginestra et al. [38], based on the particular structuring of the surfaces in this study (0° and 90° build orientation) preferential osteoblast settlement was observed in 90° specimens. This is due to the microstructuring, which has a direct effect on cell–cell interactions [44] and on the actin cytoskeleton and its remodeling, which is associated with direct mechanotransduction, leading to the increased expression of osteogenic markers such as *RUNX2*, *SPP1*, and other genes [59,60,61]. In fact, we could detect an upregulation of osteogenic genes with dependence on the build orientation but also with dependence on the niobium content. The latter also correlates with cell morphology and, accordingly, with the distribution of the actin cytoskeleton. Our results are thus also consistent with the study by Lauria et al. [62], who demonstrated that a nanoporous Ti–45Nb surface may be associated with an increased surface nanostructure, which was beneficial for MG-63 osteosarcoma cell settlement, pseudopodia formation, and osteogenic differentiation. Further, osteogenic differentiation on Ti–Nb–Ta alloy is also apparent on the open-porous lattice structures. In particular, a strong time-dependent effect is found, characterized by an increase in *RUNX2* and *COL1A1*. Considering the optimal osseointegration of implant structures, a coordinated remodeling of the surrounding bone tissue is essential, a process regulated by a balanced activity of MMPs and TIMPs. In our study, a clearly reduced impact on bone destruction is evident on the open-porous Ti–20Nb–6Ta lattice structures, as the gene expression of *MMP-1* is diminished and, at the same time, *TIMP1* mRNA is induced in a time-dependent manner. This was also evident in osteoblasts cultivated on dense specimens in 90° orientation. Thus, the results demonstrate that either surface structuring or the material composition has a beneficial effect on the MMP1/TIMP1 ratio through the absence of induced bone resorption processes. We have already observed a similar effect in a previous study where we found a clear correlation of the MMP1/TIMP1 ratio to the build orientation of Ti–6Al–4V specimens [63]. In addition to a possible surface structuring due to the build orientation, we were able to determine effects on *MMP1* and *TIMP1* gene expression depending on the niobium content. Here, a content of 35 wt. % Nb resulted in a clear increase in *MMP1* and *TIMP1* transcript levels, but without a significant difference to the other test specimens. This effect may be due to the increased deposition of type 1 collagen and related matrix remodeling processes. However, this remains to be proven in further studies.

Since the surfaces are initially settled by mesenchymal stem cells and osteoblasts after implantation, the osseointegrative process is also associated with the release of pro-inflammatory mediators leading to the differentiation of the stroma cells. Indeed, we determined an enhanced release of IL-6 within the 14 days of cultivation, in particular on Ti–Nb–Ta alloys with 90° surface orientation and open porous Ti–20Nb–6Ta test specimens compared to Ti–6Al–4V. At the same time, the release of IL-8 was clearly reduced on Ti–Nb–Ta alloys with 90° surface orientation. The higher content of IL-6 might not only be associated with inflammation since it was proven that the IL-6-dependent coupling mechanism also mediates bone formation [64]. Therefore, we assume that the release of IL-6 by human osteoblasts correlates directly with the osteogenic differentiation capacity of these cells seen in enhanced *RUNX2* and *COL1A1* gene expression. Besides bone-forming cells, macrophages are among the cells involved in the initial inflammatory response after implantation. In particular, the macrophage phenotype plays an important role in this process, significantly influencing the pro- or anti-inflammatory implant environment [65]. 

The study has some limitations. We have determined the compressive modulus of the Ti–Nb–Ta alloys and compared the results with the tensile modulus of other materials because the compressive modulus is rarely reported. However, since the Poisson’s ratio of titanium alloys is about 0.33, it can be concluded that the compressive modulus and the Young’s modulus are in a very similar range. Nevertheless, the tensile properties need to be investigated in future studies. In our present study, BSE measurements were performed to characterize the influence of the build orientation on the microstructure, and in addition, the previous results of Johannsen et al. [18] with similar Ti–Nb–Ta alloys and processing strategies were summarized and used to discuss the observed mechanical properties. However, a detailed investigation of the microstructure as a function of build orientation and processing strategy using XRD and EBSD to fully understand the deformation behavior is part of subsequent studies. Overall, it is not possible to conclude from this study whether the amount of niobium significantly influences biocompatibility and osteogenic differentiation, as there were only minor differences in cell behavior. Rather, build orientation seems to influence cell behavior through the structuring of the surface. In addition to the build orientation, the surface properties are influenced by laser power, scanning speed, and hatch distance. Therefore, future studies could optimize the surface properties by systematically analyzing the influence of the process strategy on the surface.

Despite these limitations, the results obtained have provided further insight into the essential properties of additively manufactured Ti–Nb–Ta alloys and have also identified further research questions. These include a comprehensive analysis of mechanical (tensile) properties as a function of build orientation, taking into account the detailed investigation of the microstructure. The influence of annealing on additively manufactured Ti–Nb–Ta has not been investigated, but it may be interesting in terms of structural changes and their influence on mechanical properties. Furthermore, the influence of differently structured surfaces on cell interaction, the specific macrophage response to Ti–Nb–Ta alloys, the surface properties in terms of the characterization of passive oxide layers, e.g., by XPS, and metal ion release in physiological solutions should be characterized in future studies.

## 5. Conclusions

Ti–xNb–6Ta (x = 20, 27, 35) alloys are promising materials for use in additively manufactured bone implants. The mechanical properties of Ti–Nb–Ta specimens produced by PBF-LB/M are controlled by the chemical composition of the alloy and the orientation of the specimen in the build chamber. Ti–Nb–Ta alloys have lower compressive moduli (Ti–20Nb–6Ta, 90°: 43 GPa) compared to commonly used Ti–6Al–4V (~114 GPa) implant materials, which reduces the mechanical mismatch with human bone and, thus, suppresses the tendency for stress shielding. The elasticity can be further reduced by fabricating open porous specimens (Ti–20Nb–6Ta: 8.7 GPa), thereby minimizing the mechanical mismatch with bone tissue. The biological properties were mainly influenced by the build orientation and cultivation time. As-printed surfaces of the Ti–Nb–Ta alloys promoted improved osteogenesis and reduced inflammation compared to Ti–6Al–4V. In conclusion, Ti–Nb–Ta alloys show great potential for use in orthopedic and dental implants, outperforming commonly used implant materials such as Ti–6Al–4V in the aspects investigated.

## Figures and Tables

**Figure 1 jfb-15-00046-f001:**
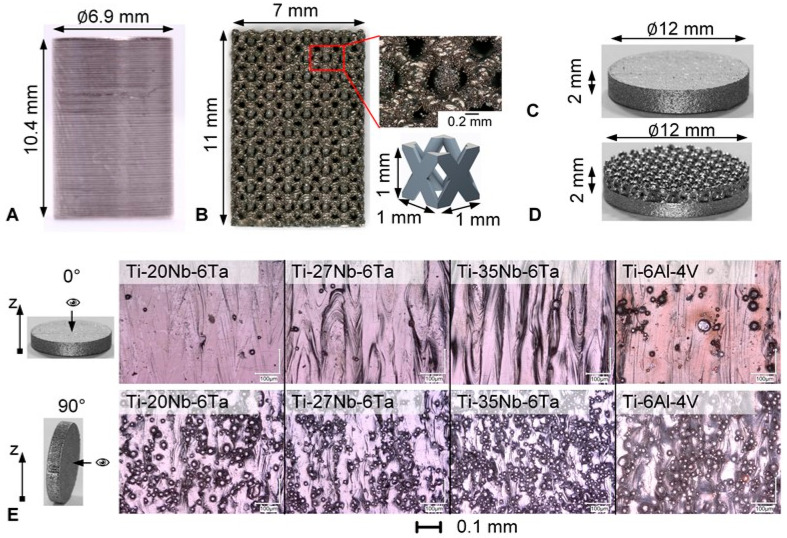
Overview of specimens manufactured by laser beam powder bed fusion for mechanical and biological characterization. (**A**) Machined, dense cylindrical specimens for mechanical characterization, (**B**) rectangular open porous lattice-structured specimen for mechanical characterization, magnified view of the structure and the CAD design of the face-centered cubic unit cell, (**C**) cylindrical dense specimens for biological characterization, (**D**) open porous lattice-structured specimen for biological characterization with one layer of the defined unit cells, and (**E**) digital microscopic images (200-times magnification) of as-printed surfaces in 0° and 90° build orientation (z indicating build direction) of the Ti–xNb–6Ta (x = 20, 27, 35) and Ti–6Al–4V specimens.

**Figure 2 jfb-15-00046-f002:**
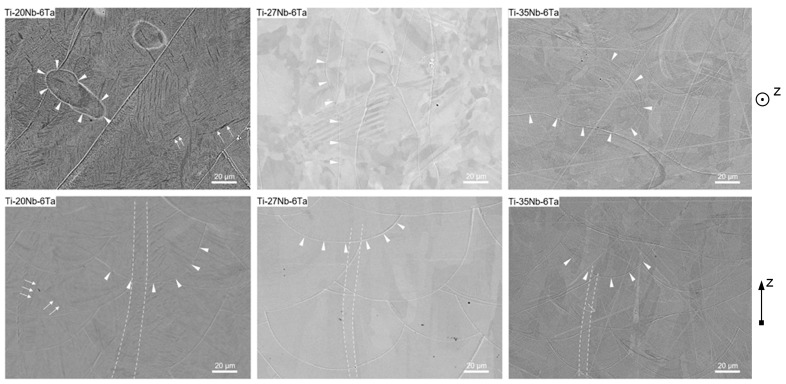
Backscattered electron analysis (BSE) of Ti–20Nb–6Ta, Ti–27Nb–6Ta, and Ti–35Nb–6Ta additively built in 0° (top) and 90° direction (bottom). Platelet-like structures (only Ti–20Nb–6Ta), elongated (columnar) grains (in 90°) and melt pool boundaries are highlighted by white arrows, dotted lines, and white triangles, respectively. The BSE images of Ti–35Nb–6Ta also feature scratches, which could not be removed by polishing.

**Figure 3 jfb-15-00046-f003:**
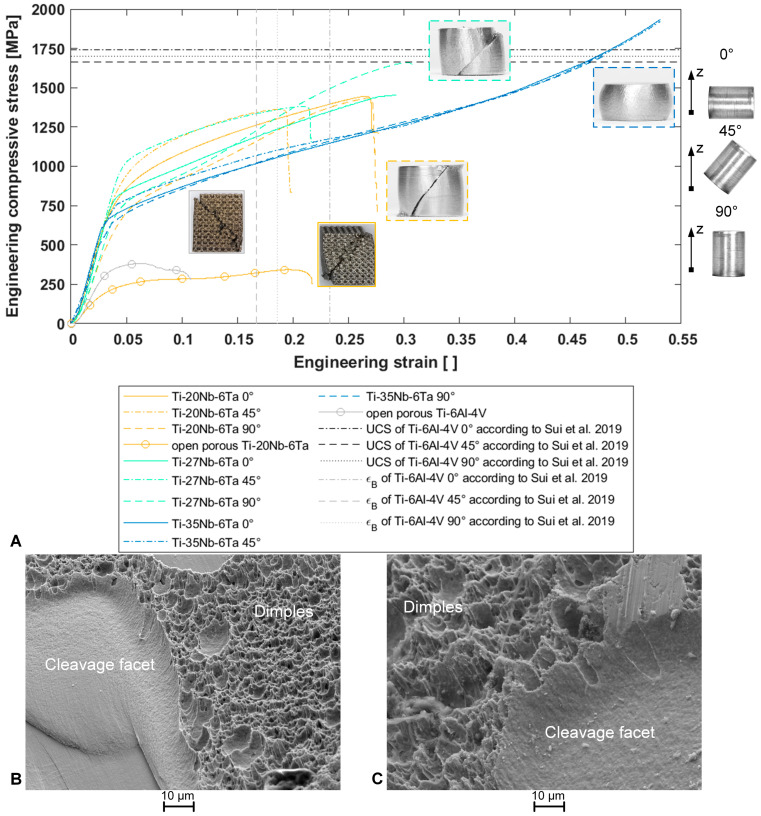
(**A**) Representative stress–strain curves and fracture images of dense Ti–xNb–6Ta (x = 20, 27, 35) specimens in different build orientations (0°, 45°, 90°) and open porous lattice structured Ti–20Nb–6Ta and Ti–6Al–4V specimens, where all specimens were manufactured by laser beam powder bed fusion of pre-alloyed spherical powders. Ultimate compressive strength (UCS) and elongation at break (ε_B_) of Ti–6Al–4V in different build orientations according to Sui et al., 2019 [31] are shown for comparison, and the right image illustrates the orientation of the specimens in the build chamber (z: build direction). (**B**,**C**) Exemplary fracture images obtained by field emission scanning electron microscopy of the fracture surfaces at 1000× magnification of dense Ti–20Nb–6Ta specimens in (**B**) 0° and (**C**) 45° build orientation.

**Figure 4 jfb-15-00046-f004:**
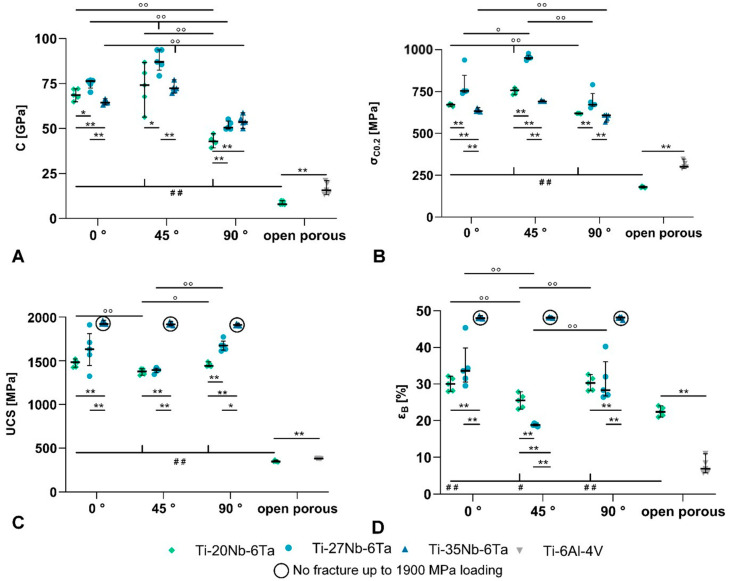
Measured compressive mechanical properties. (**A**) C: compression module, (**B**) σ_C0.2_: compressive yield strength, (**C**) UCS: ultimate compressive strength, and (**D**) ε_B_: elongation at break of the dense Ti–xNb–6Ta (x = 20, 27, 35) alloys additively manufactured by laser beam powder bed fusion in 0°, 45°, and 90° build orientation as well as the open porous lattice-structured Ti–20Nb–6Ta and Ti–6Al–4Vspecimens. Results of the pairwise Mann–Whitney U tests (n = 5) are indicated as follows: ° *p* < 0.05, °° *p* < 0.01 (effect of build direction); * *p* < 0.05, ** *p* < 0.01 (effect of material); # *p* < 0.05, ## *p* < 0.01 (effect of specimen structure).

**Figure 5 jfb-15-00046-f005:**
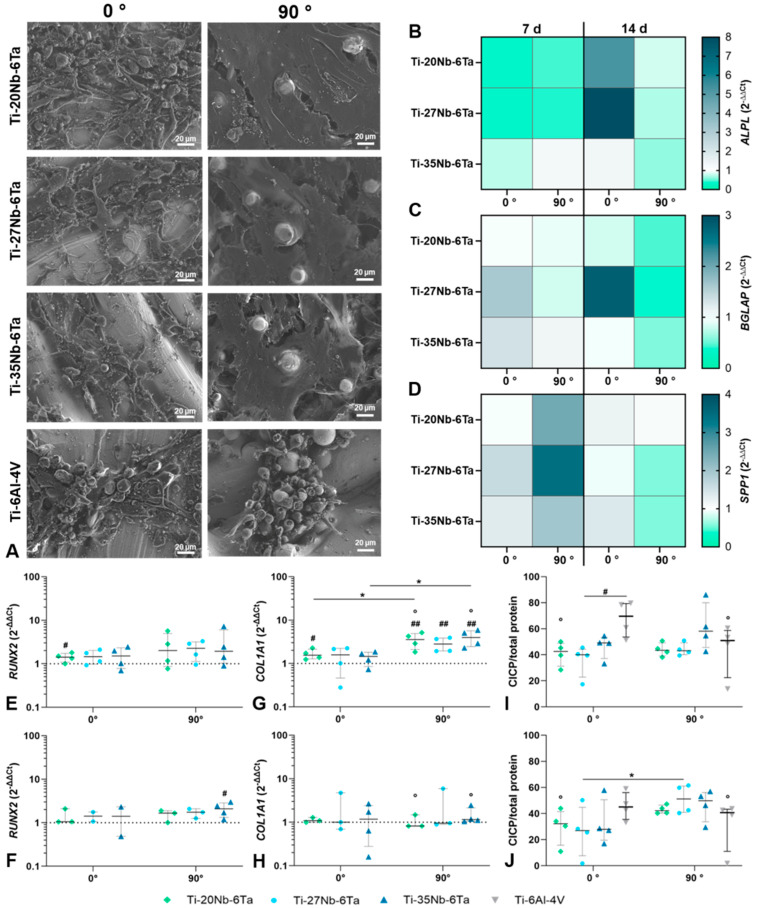
Biological characterization of the dense Ti–xNb–6Ta (x = 20, 27, 35) or Ti–6Al–4V specimens under consideration of cell spreading and induction of osteoblastic differentiation. (**A**) The population of the surfaces of the specimens with 0° and 90° build orientation with human osteoblasts was determined after seven days via field emission scanning electron microscopy. Bar corresponds to 20 µm. (**B**–**J**) The differentiation capacity of human osteoblasts on the different surfaces of specimens was determined via gene expression analyses and quantification of CICP after seven and fourteen days. (**B**–**D**) Heatmaps of ALPL, BGLAP, and SPP1 gene expression. Color coding corresponds to the median of at least two independent pre-osteoblast donors, with a white background corresponding to the gene expression level of osteoblasts on Ti-6Al-4V (“1”). A downregulated gene expression rate is displayed in green, with “0” corresponding to only low expression. Conversely, up-regulation is highlighted with a blue color code. (**E**–**H**) Gene expression levels of RUNX2 and COL1A1 after seven (**E**,**G**) and fourteen days (**F**,**H**). Data are depicted as individual values with median, minimum, and maximum (n ≥ 2). Gene expression rates of osteoblasts on Ti–6Al–4V specimens served as controls (dotted line at “1”). (**I**,**J**) Protein amount of cross-linked C-telopeptides of Type 1 collagen (CICP) after seven (**I**) and fourteen days (**J**) related to total protein (ng/mg, n = 4). Statistical significance was determined via unpaired (gene expression) and paired (protein quantification) *t*-test with * *p* < 0.05 (effect of build orientation); # *p* < 0.05, ## *p* < 0.01 (effect of the Ti alloy); ° *p* < 0.05 (effect of cultivation time).

**Figure 6 jfb-15-00046-f006:**
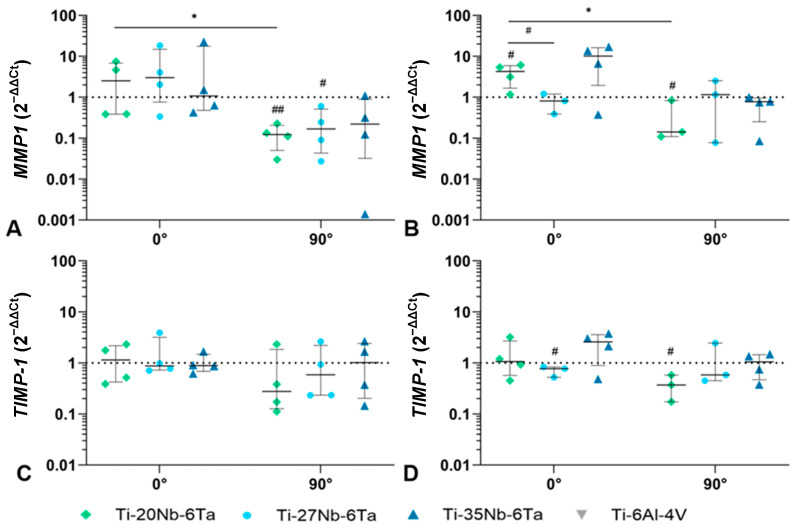
Gene expression analysis of MMP1 (**A**,**B**) and TIMP1 (**C**,**D**) of osteoblasts cultivated on dense Ti–xNb–6Ta (x = 20, 27, 35) or Ti–6Al–4V specimens after 7 (**A**,**C**) and 14 days (**B**,**D**). Data are depicted as individual values with median, minimum, and maximum (n ≥ 3). Statistical significance was determined by an unpaired *t*-test with * *p* < 0.05 (effect of build orientation); # *p* < 0.05, ## *p* < 0.01 (effect of the material).

**Figure 7 jfb-15-00046-f007:**
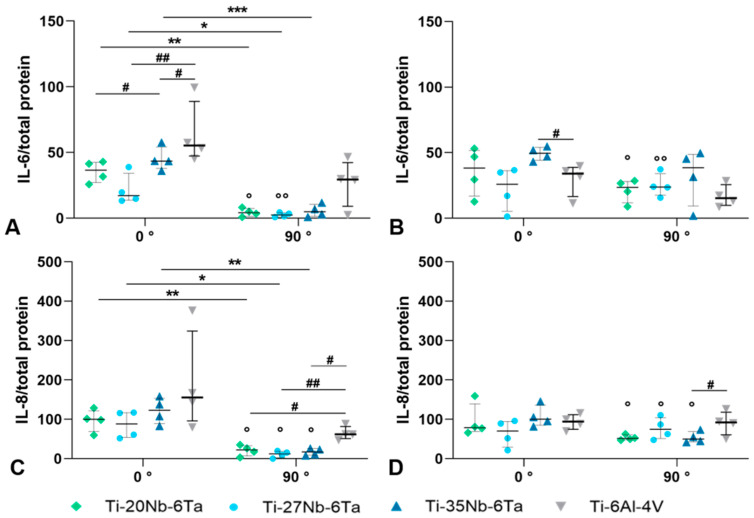
Induction of inflammation in human osteoblasts seeded on Ti–xNb–6Ta (x = 20, 27, 35) or Ti–6Al–4V specimens. The release of interleukin (IL-) 6 (**A**,**B**) and IL-8 (**C**,**D**) was determined via ELISA after 7 (**A**,**C**) and 14 days (**B**,**D**). Protein amounts were related to total protein (pg × mg^−1^, n = 4). Statistical significance was determined by a paired *t*-test with * *p* < 0.05, ** *p* < 0.01, *** *p* < 0.001 (effect of build orientation); # *p* < 0.05, ## *p* < 0.01 (effect of the material); ° *p* < 0.05, °° *p* < 0.01 (effect of cultivation time).

**Figure 8 jfb-15-00046-f008:**
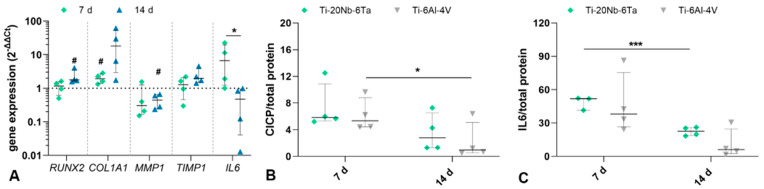
Biological characterization of open porous lattice-structured specimens (Ti–20Nb–6Ta vs. Ti–6Al–4V). The differentiation capacity, the induction of inflammation, and the induction of bone remodeling in human osteoblasts cultivated on open porous specimens were determined via gene expression analyses and protein quantification after 7 and 14 days. (**A**) Gene expression levels of RUNX2, COL1A1, IL6, MMP1, and TIMP1 are depicted as individual values with median, minimum, and maximum (n = 4). Gene expression rates of osteoblasts on Ti–6Al–4V specimens served as controls (dotted line at “1”). Protein amount of (**B**) cross-linked C-telopeptides of Type 1 collagen (CICP in ng/mg; n ≥ 3) and (**C**) interleukin 6 (IL-6 in pg × mg^−1^) related to total protein. Statistical significance was determined by a paired (gene expression, CICP) and an unpaired (IL-6) *t*-test with * *p* < 0.05, *** *p* < 0.001 (effect of cultivation time); # *p* < 0.05 (effect of the material).

**Table 1 jfb-15-00046-t001:** Process parameters and resulting density during laser beam powder bed fusion of Ti–xNb–6Ta (x = 20, 27, 35) alloys and Ti–6Al–4V specimens and the resulting relative density shown as mean and standard deviation (in brackets).

Material	Scanning Speed [mm/s]	Laser Power [W]	Layer Thickness [mm]	Resulting Density [%]	
Ti–20Nb–6Ta	1250	170	0.30	99.96	(0.01) ^a^
Ti–27Nb–6Ta	1350	170	0.30	99.97	(0.01) ^a^
Ti–35Nb–6Ta	1500	170	0.30	99.97	(0.01) ^a^
Ti–6Al–4V	1200	240	0.60	>99.7 ^b^	

^a^ Density analysis of Ti–Nb–Ta alloys conducted by Johannsen et al. [18] with similar processing parameters and Ti–Nb–Ta powders. ^b^ Not specifically measured for this study, but typical value of the established process at the Fraunhofer Research Institution for Additive Manufacturing Technologies IAPT, Hamburg, Germany.

**Table 2 jfb-15-00046-t002:** Measured dimensions as mean value and standard deviation (in brackets) of the strut thickness and pore size (each in frontal and top view) as well as the porosity measured by µCT scans of the open porous lattice–structured Ti–20Nb–6Ta and Ti–6Al–4V specimens.

Material	Strut Thickness [mm]				Pore Size [mm]				Porosity [%]	
	Front		Top		Front		Top			
Ti–20Nb–6Ta	0.47	(0.04)	0.43	(0.05)	0.47	(0.06)	0.54	(0.05)	49.47	(0.49)
Ti–6Al–4V	0.44	(0.02)	0.46	(0.02)	0.58	(0.02)	0.55	(0.02)	47.36	(1.30)

**Table 3 jfb-15-00046-t003:** Measured surface roughness (Ra: average roughness, Rz: arithmetical mean deviation of the profile) of the specimens for biological characterization in 0° and 90° build orientation (BO) shown as mean and standard deviation (in brackets).

Material	Ti–20Nb–6Ta				Ti–27Nb–6Ta				Ti–35Nb–6Ta				Ti–6Al–4V			
BO [°]	0		90		0		90		0		90		0		90	
Ra [µm]	4.0	(0.7)	5.7	(0.8)	6.6	(1.3)	8.0	(2.0)	9.0	(1.9)	8.6	(1.6)	5.9	(0.9)	11.6	(1.0)
Rz [µm]	34.7	(8.1)	47.9	(5.2)	56.2	(13.7)	63.5	(9.7)	77.1	(16.3)	68.9	(9.8)	68.1	(11.7)	103.5	(18.3)

**Table 4 jfb-15-00046-t004:** Overview of the genes of interest and their primer sequences.

Primer	Sequences (5′–3′)
Actin Beta (*ACTB*)	fwd: CTTCCTGGGCATGGAGTC rev: AGCACTGTGTTGGCGTACAG
Alkaline Phosphatase, Biomineralization Associated (*ALPL*)	fwd: CATTGTGACCACCACGAGAG rev: CCATGATCACGTCAATGTCC
Bone Gamma-Carboxyglutamate Protein (*BGLAP*)	fwd: TCAGCCAACTCGTCACAGTC rev: GGTGCAGCCTTTGTGTCC
Collagen Type I Alpha 1 Chain (*COL1A1*)	fwd: ACGAAGACATCCCACCAATC rev: AGATCACGTCATCGCACAAC
Interleukin 6 (*IL6*)	fwd: TGGATTCAATGAGGAGACTTGCC rev: CTGGCATTTGTGGTTGGGTC
Matrix Metallopeptidase 1 (*MMP1*)	fwd: AGAGCAGATGTGGACCATGC rev: TCCCGATGATCTCCCCTGAC
Runt-related Transcription Factor 2 (*RUNX2*)	fwd: CGCCTCACAAACAACCACAG rev: ACTGCTTGCAGCCTTAAATGAC
Secreted Phosphoprotein 1, Osteopontin (*SPP1*)	fwd: AACGCCGACCAAGGAAAACT rev: GCACAGGTGATGCCTAGGAG
Secreted protein acidic and rich in cysteine, Osteonectin (*SPARC*)	fwd: CTGGACTACATCGGGCCTTG rev: ATGGATCTTCTTCACCCGCAG
Tissue inhibitor of metalloproteinase-1 (*TIMP1*)	fwd: ATTGCTGGAAAACTGCAGGATG rev: GTCCACAAGCAATGAGTGCC

**Table 5 jfb-15-00046-t005:** Mechanical properties (C: compression module, σ_C0.2_: compressive yield strength, UCS: ultimate compressive strength, ε_B_: elongation at break, EAS: elastic admissible strain as ratio of σ_C0.2_ to C) of the dense Ti–xNb–6Ta (x = 20, 27, 35) alloys additively manufactured by laser beam powder bed fusion in 0°, 45°, and 90° build orientation (BO) as well as of open porous lattice-structured Ti–20Nb–6Ta and Ti–6Al–4V specimens.

Nb [wt. %]	BO [°]	C [GPa]		σ_C0.2_ [MPa]		UCS [MPa]		ε_B_ [%]		EAS [ ]	
Densely manufactured											
20	0	69.0	(2.7)	670.1	(6.3)	1467.8	(37.1)	29.9	(1.9)	0.97	(0.03)
	45	73.2	(10.6)	751.8	(15.6)	1374.6	(30.2)	25.3	(1.7)	1.05	(0.17)
	90	43.1	(2.6)	619.1	(2.0)	1453.3	(21.0)	30.2	(1.7)	1.44	(0.09)
27	0	74.9	(2.5)	787.3	(76.1)	1629.1	(191.8)	34.9	(5.5)	1.05	(0.11)
	45	87.9	(5.4)	954.4	(13.6)	1389.6	(23.9)	18.8	0.3)	1.09	(0.07)
	90	51.7	(2.1)	695.1	(49.1)	1673.6	(55.7)	30.8	(5.1)	1.35	(0.11)
35 ^a^	0	64.5	(1.1)	636.7	(8.2)	-		48.1	(0.3)	0.99	(0.02)
	45	72.6	(2.5)	693.4	(3.1)	-		48.1	(0.1)	0.96	(0.03)
	90	54.2	(2.9)	597.2	(17.0)	-		48.1	(0.3)	1.12	(0.02)
Open porous lattice-structured											
Ti–20Nb–6Ta		8.7	(1.0)	179.2	(4.5)	351.7	(7.5)	22.5	(1.0)	0.48	(0.07)
Ti–6Al–4V		16.7	(3.2)	312.0	(20.0)	381.7	(2.0)	7.63	(1.9)	0.53	(0.08)

^a^ No fracture up to ~1900 MPa compressive stress applied.

## Data Availability

The raw data supporting the conclusions of this article will be made available by the authors on request.

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
