# Peer review of "Advanced Ti–Nb–Ta Alloys for Bone Implants with Improved Functionality"

_jfb, 2024, doi:10.3390/jfb15020046_

Round 1

Reviewer 1 Report

Comments and Suggestions for Authors

The manuscript by Sass et al. reports a study on Titanium-Niobium-Tantalum alloys manufactured by laser beam powder bed fusion. The investigated 3D metal printing process enables the production of components with a high degree of freedom in design. In a previous study (https://doi.org/10.1016/j.matdes.2023.112265) the authors reported the development and microstructure characterization of the investigated system. The present study is addressed to an in-depth mechanical and biological characterization of the material.

The paper is well written and the results and discussion sections are clear and detailed.  The references are appropriate and the quality of tables and figures is fine.The conclusions are consistent with the results presented.

I would like only to suggest as further development to gain insight into chemical characterization of the materials (i.e. surface characterization by XPS, ion release studies, etc.).

Text editing: Please check the manuscript since some typing errors are present (i.e. properties at line 62, Ti-6Al-4V at lines 37 and 40, etc.).

Reviewer 2 Report

Comments and Suggestions for Authors

1."Are these three directional samples manufactured directly or cut from a single piece of a sample?

2. Have all samples been sandblasted? Or some other surface treatment? How do the authors view the effects of surface residual powder on biological properties?

3. The mechanical properties of the samples in the three directions are different, and the reasons for this need to be explained in more detail. The following article may help explain this problem: https://doi.org/10.1002/advs.202302884

4. The influence of Nb content on elastic modulus and mechanical properties of alloys has been studied extensively. What's new about the conclusion of this article?

5. Although the amount of data is large, the content expressed in the picture is unclear, and the picture itself is not clear enough. Suggested modification."

Comments on the Quality of English Language

The quality of the English language in the article is good, but some minor errors require editing. For instance, in Table 5, "Ti20-Nb-6Ta" has been misspelled. Therefore, it is recommended to thoroughly review and correct these errors before publishing the article.

Reviewer 3 Report

Comments and Suggestions for Authors

The article attempts to analyse and determine the bio-functionalization of different variants of Ti-Nb-Ta. The work comprises two parts: mechanical and biological characterization. The biological characterization shows several experiments with very interesting results, while the mechanical one lacks information. Furthermore, the authors cannot explain and justify the effects of the material (chemical composition) and building direction on the biological response.

The authors must consider the comments below and re-submit the manuscript for a second round of evaluation.

Comments:

+ In the section Introduction, the authors must describe more deeply works related to similar systems already done by additive manufacturing. Reference 18, from part of the co-authors, should be developed to point out those missing points, to be expected to be covered in the current work.

+ Materials and Methods: In general, the description of the experimental activities is precise. However, the authors should clarify if the Ti-Nb-Ta alloys were heat treated after the building because it is important for further understanding the results. In the preparation of samples for biological characterization, the authors must justify the use of samples with different roughness since, in general, the surface is prepared with the same roughness (e.g., use of SiC 4000). They name this effect in the discussion and it makes it hard to compare the materials seriously (Page 17, Lines 585-586)

+ Results: the authors must show the microstructure of the different alloys and determine the influence on the compression tests. If the work is based on reference 18 (Johannsen et. al), all the materials are chemically heterogeneous, and it may affect the mechanical and biological properties.

++ Could you explain why one alloy did not show fracture or damage at the surface?

++ How reliable is the compression module? Why are these values compared to the literature when most of the information is E?

+ SEM, EDS/WDS, EBSD and LOM must be shown and explained. It will help to improve the discussion of the work. The authors assume much information, but they need to confirm the observation by the samples used in the work. There is no justification for the surface and printing strategy on the biological response. The authors try to skip this analysis (page 17, lines 554-555), but I am finding very important for the quality of the work.

+ A deep analysis of biological results concerning the Nb content is missing.

+ Conclusions need to give more substantial information.

Reviewer 4 Report

Comments and Suggestions for Authors

The paper presents a detailed investigation of a tertiary ti-based alloy for biomedical applications. The paper is well written and the results are clearly presented. There are just minor comments to be addressed. What curve, among the 5 replications, is shown in Figure 2A? The average value or one of the five?

Comments on the Quality of English Language

Minor editing needed 

Reviewer 5 Report

Comments and Suggestions for Authors

The manuscript titled Advanced Ti-Nb-Ta alloys for bone implants with improved functionality aims to assess the mechanical and biological properties of Ti-xNb-6Ta (x = 20, 27, 35) alloys manufactured by PBF-LB/M. The influence of the build orientation during manufacturing on different properties was evaluated. In addition to fully dense specimens, open porous Ti-20Nb-6Ta specimens with a defined lattice structure were also characterized and compared to Ti-6Al-4V specimens. 

It is well written, the results are clearly displayed and the discussion is well conducted.

My comments and suggestions for a few minor revisions are as follows:

The manuscript has to be further checked for typing/editing errors.

Line 67. In my opinion, the reference citation [18], is not appropriate when describing the aim of your study. 

The comparison with the Ti-6Al-4V should be added in the aim of the study. 

Line 76. In my opinion Titanium-niobium-tantalum is not necessary, Ti-Nb-Ta would be enough, as you have used this format before. 

Line 80. In my opinion, the composition should be displayed, maybe the reader does not wish to access reference 18 for more data.

Comments on the Quality of English Language

Minor editing and typing errors should be corrected. 

Round 2

Reviewer 3 Report

Comments and Suggestions for Authors

Page 20, Lines 613 and 614. Modify the text as follows:

…[18], it is expected similar microstructure in the present work. , instead of  [18], it is obvious that identical microstructures were obtained compared to those published.